# No association between autistic traits and contextual influences on eye-movements during reading

Nathan Caruana[1,2] and Jon Brock[1,2,3]

[1] ARC Centre of Excellence in Cognition and its Disorders, Australia[*]
[2] Department of Cognitive Science, Macquarie University, Sydney, Australia
[3] Department of Psychology, Macquarie University, Sydney, Australia

## ABSTRACT

Individuals with autism spectrum disorders are claimed to show a local cognitive bias, termed "weak central coherence", which manifests in a reduced influence of contextual information on linguistic processing. Here, we investigated whether this bias might also be demonstrated by individuals who exhibit sub-clinical levels of autistic traits, as has been found for other aspects of autistic cognition. The eye-movements of 71 university students were monitored as they completed a reading comprehension task. Consistent with previous studies, participants made shorter fixations on words that were highly predicted on the basis of preceding sentence context. However, contrary to the weak central coherence account, this effect was not reduced amongst individuals with high levels of autistic traits, as measured by the Autism Spectrum Quotient (AQ). Further exploratory analyses revealed that participants with high AQ scores fixated longer on words that resolved the meaning of an earlier homograph. However, this was only the case for sentences where the two potential meanings of the homograph result in different pronunciations. The results provide tentative evidence for differences in reading style that are associated with autistic traits, but fail to support the notion of weak central coherence extending into the non-autistic population.

Corresponding author
Jon Brock, jon.brock@mq.edu.au

## INTRODUCTION

Autism spectrum disorders are currently defined and diagnosed in terms of clinically significant social and communication impairments, co-occurring with repetitive behaviours and restricted interests (*APA, 2013*). Diagnosis is categorical but it is generally acknowledged that there is no clear cut off, with autistic-like behavioural traits being continuously distributed in the general population. Moreover, a number of studies have reported that non-autistic individuals who self-report high levels of autistic traits also evidence cognitive strengths and weaknesses that are similar to those identified in studies

* www.ccd.edu.au

of individuals with a clinical diagnosis of autism. Examples include impaired performance on a test of facial emotion recognition (*Baron-Cohen et al., 2001*; *Voracek & Dressler, 2006*) and enhanced performance on visual search tasks (*Almeida et al., 2010*; *Brock, Xu & Brooks, 2011*; *Milne et al., 2013*; but see *Gregory & Plaisted-Grant, in press*).

The current study was motivated by another classic finding in autism research—the poor performance of autistic individuals on a test of homograph reading (see *Brock & Caruana, 2014* for review). In the homograph reading test, participants read aloud sentences containing heterophonic homographs—words such as "tear" and "bow" that have two or more meanings associated with different pronunciations. If the sentence has been understood correctly then participants should give the contextually appropriate pronunciation of the homograph. However, autistic individuals tend to perform relatively poorly on the test, suggesting a failure of sentence-level language comprehension (*Burnette et al., 2005*; *Frith & Snowling, 1983*; *Happé, 1997*; *Jolliffe & Baron-Cohen, 1999*; *Lopez & Leekam, 2003*; but see *Snowling & Frith, 1986*).

Impaired homograph reading has been interpreted in terms of a deficit in context processing, termed "weak central coherence" (*Frith, 1989*). On this view, autistic individuals make errors on the task because they process each word in isolation, ignoring the surrounding context. However, studies involving ambiguous *spoken* words have been less supportive of this account, indicating that individuals with autism show a degree of sensitivity to sentence context that is commensurate with their language abilities (*Brock et al., 2008*; *Henderson, Clarke & Snowling, 2011*; *Lopez & Leekam, 2003*; *Norbury, 2005*). For example, *Brock et al. (2008)* used a language-mediated eye-movements paradigm in which participants viewed a display of four objects whilst listening to spoken sentences. Children with autism and control children matched on language ability showed the same tendency to make anticipatory saccades towards objects that were predicted by the sentence context. They also showed the same mediating effect of sentence context on gaze towards objects that were phonologically similar to the word they were hearing. These findings challenge the central coherence account and suggest that there may be some alternative explanation for poor performance on the homograph test.

In their original study of homograph reading, *Frith & Snowling (1983)* noted that, whereas typically developing and dyslexic children often hesitated or began the sentence again after they had mispronounced a homograph, autistic children "never showed any signs of being aware of their errors" (p. 336). Similarly, *Happé (1997)* noticed "striking" differences in the tendency of autistic and non-autistic participants to self-correct their homograph reading errors. Such observations suggest that poor performance may reflect, not a failure of context sensitivity, but a failure of comprehension monitoring. That is, when autistic individuals misconstrue the homograph, they may not subsequently recognize when the sentence has stopped making sense.

In fact, *Happé (1997)* argued against this comprehension monitoring account, noting that group differences in performance remained when self-corrections were ignored and participants were scored only on their initial attempts at producing the homograph (see also *Lopez & Leekam, 2003*). However, this argument rests on the assumption that the

participant's first attempt at articulating the homograph necessarily corresponds to their initial *interpretation* of it. This is clearly not the case, as many participants perform the task without overt errors, even when the disambiguation comes some time *after* the homograph. Indeed, a recent eye-tracking study of the task showed a considerable lag between participants fixating on the homograph and beginning to articulate it (*Brock & Bzishvili, 2013*). Given the challenges to the weak central coherence account, the issue of comprehension monitoring in autism is certainly worth revisiting.

Differentiating between these two opposing accounts of homograph reading impairments is difficult using the task itself. Thus, a new paradigm is called for. As a forerunner to studies of individuals with autism, the current study aimed to contrast these two opposing accounts of impaired homograph reading by looking at the relationship between autistic traits in a nonclinical population and participants' eye-movements during reading.

To test the "central coherence" account, participants read a series of short sentences involving a predictability manipulation, whereby the same target words were either highly predictable or completely unpredictable (although not semantically anomalous) based on the preceding sentence stem. Previous research has shown that readers spend less time fixating on words the more predictable they are (*Ehrlich & Rayner, 1981*), presumably because the processing of words is facilitated if they are already anticipated (*Kliegl, Nuthmann & Engbert, 2006*; *Rayner & Well, 1996*). If individuals with autism process words *out* of context, we would expect this contextual facilitation effect to be reduced amongst those reporting high levels of autism-like traits.

The "comprehension monitoring" account was assessed via an ambiguity manipulation. Participants read sentences containing an early homograph that was later disambiguated towards its less common meaning. In a corresponding control condition, the same sentences were presented but with the homograph replaced by an unambiguous synonym. Previous studies have shown that participants spend longer reading regions of text that disambiguate an earlier homograph (*Duffy, Morris & Rayner, 1988*; *Rayner & Duffy, 1986*). This is attributed to the longer time required to integrate the disambiguating word with the preceding sentence, particularly if it requires a reevaluation of the meaning of the homograph. However, *van der Schoot et al. (2009)* found that non-autistic children with reading comprehension difficulties failed to show an ambiguity effect in this paradigm. This suggests that these children were unaware when they had misinterpreted the homograph and thus made no attempt to reconcile the disambiguating word with the homograph. As the authors noted, this finding is consistent with a large body of evidence for reduced comprehension monitoring in this population (cf. *Ehrlich, 1996*; *Ehrlich, Remond & Tardieu, 1999*; *Yuill & Oakhill, 1991*; *Zubrucky & Moore, 1989*). If individuals with autism also have difficulties in comprehension monitoring, we would likewise expect a reduction in this ambiguity effect amongst participants with high levels of autistic traits.

## METHOD

### Ethics

The study was approved by the Macquarie University Human Research Ethics Committee (Ref D00167). Participants provided written consent prior to participation.

### Participants

Seventy-one 18- to 23-year-old undergraduate students (49 females, 22 males) were recruited at Macquarie University, Sydney where they received course credit for their participation. All participants were native English speakers and had either normal or corrected to normal vision.

The *Autism Spectrum Quotient* (AQ; *Baron-Cohen et al., 2001*) was used as a measure of sub-clinical autistic traits. This is a 50-item questionnaire organized into five domains—social skills, attention switching, attention to detail, communication and imagination. It has high test-retest reliability ($r = .7$, $p = .002$; *Baron-Cohen et al., 2001*) and provides good discrimination between high functioning individuals with autism and other clinical and non-clinical groups (*Baron-Cohen et al., 2001*; *Hoekstra et al., 2008*). Our participants' scores ranged from 4 to 28 (mean = 14.9, SD = 5.1).

Previous studies of homograph reading in autism have matched participant groups on receptive vocabulary knowledge. Here, we used the *vocabulary* scale of the standardized Shipley-2 Composite A as a measure of written word knowledge (*Shipley et al., 2009*) and a potential covariate in analyses of eye-movements. The scale consists of 40 multiple-choice items in which individuals select the appropriate synonym for a target word (e.g., PARDON) from four alternatives (e.g., forgive, pound, divide, or crash). Participants' scores ranged from 20 to 38 (mean = 30.0, SD = 3.6).

### Stimuli

Stimuli for the predictability manipulation (see Supplemental Information) were adapted from the Speech Perception in Noise stimulus set (*Kalikow, Stevens & Elliott, 1977*) in which the same words appear at the end of two sentences—one that is highly constraining and one in which there is essentially no constraint provided by the preceding context. Our adaptations involved adding extra words to the end of each sentence so that the target word was not the final word (see Supplemental Information for complete sentence sets).

(1) Crocodiles live in muddy *swamps* most of the time.
(2) The girl knows about the *swamps* in the bush.

For the ambiguity manipulation (Supplemental Information), we first identified 30 noun–noun or verb–verb homographs, including 25 homophonous (same pronunciation for both meanings) and 5 heterophonic (different pronunciation) pairs. From these, we created 30 sentences in which the meaning of the homograph early in the sentence could be altered by changing a single word later in the sentence. A sentence stem completion task was administered to 45 Macquarie University students (not participants in the main

experiment), who were asked to read the 30 sentence stems (3) and complete each sentence using the first word that came to mind.

(3) The crane was slowly ————.

We then calculated for each sentence stem the proportion of responses that were consistent with each of the possible meanings of the homograph (disregarding any ambiguous or nonsensical responses) and chose the less common meaning, adding extra words after the disambiguating word (4). Thirty matched unambiguous sentences were also constructed by replacing the homograph with an unambiguous word that was semantically related to the less common meaning of the homograph (5).

(4) The crane was slowly flying over the lake.
(5) The bird was slowly flying over the lake.

The stimuli from the predictability and ambiguity manipulations were divided into two lists, each consisting of 55 sentences, such that (a) the number of predictable, unpredictable, ambiguous, and unambiguous sentences was balanced across alternate forms; and (b) members of sentence pairs (e.g., (1) and (2); (4) and (5)) were assigned to different forms. Half the participants received one form and half the other, although each participant was presented with sentences in a different random order.

## Apparatus

Participants were seated approximately 70 cm away from a 40 cm × 40 cm display screen. The right eye was tracked at 500 Hz using an Eyelink 1,000 remote eye tracker. The system was mounted below the desktop display in front of the participant, and consisted of a camera and infrared illuminator. Participants were required to wear a small circular target sticker on their forehead, allowing them to move freely within a 20 cm radius during the experiment. A standard (for reading experiments) three-point camera calibration and validation was conducted prior to the test phase with the three points in a horizontal row at the same screen height as the text.

## Procedure

Stimuli were presented using the SR Research Experiment Builder software (*SR Research, 2004*). Participants were instructed to silently read each sentence and press the space bar to indicate that they were ready for the next sentence. Participants were also informed that after some trials, the sentence would be followed by a related comprehension question. This ensured that they were reading for meaning, and were appropriately attending to the stimuli. Four practice trials were conducted before the test phase. Although no feedback was provided, participants had the opportunity to ask questions before beginning the test trials.

Each trial began with a fixation point at the left of the screen. When the participant was looking at the fixation point, the experimenter would cue the sentence, with the first word appearing at the fixation point location. Comprehension questions followed 40% of trials

(see Supplemental Information). Participants gave a yes or no response by pressing the "Y" or "N" keys respectively. Unfortunately, these responses were not recorded due to a programming error.

Following the eye-tracking tasks, subjects completed the vocabulary test and finally the AQ.

## Data screening

For the predictability manipulation, we required that the first fixation on the target word was progressive (i.e., it was not preceded by a fixation on a word later in the sentence), and lasted at least 50 ms (*Rayner, 2009*). In total, there were 1,446 valid trials (81.5%).

For the ambiguity manipulation, we required a valid fixation on the disambiguating word (using the same criteria as above). A further criterion was that the homograph (or control word) was fixated before the disambiguating word. Screening left 1,762 trials (82.7%) for analysis.

## Statistical analyses

Analyses focused on first fixation duration on the relevant word and first run duration (the sum of consecutive fixations on the same word). Durations were log-transformed (cf. *Hohenstein, Laubrock & Kliegl, 2010*) and subjected to mixed random effects analyses (*Baayen, Davidson & Bates, 2008*) using the lme4 (lme4_0.999375-42) library (*Bates, 2005*) in R (2.13.0; *R Development Core Team, 2012*). In all analyses, condition (predictable vs. unpredictable; homograph vs. unambiguous) was treated as a binary fixed factor, coded as $\pm 0.5$. For the ambiguity manipulation, homophony (of the homograph in the pair) was also coded as a fixed factor, but because there were more homophonic than heterophonic homographs, they were coded as $+0.1667$ and $-0.8333$ so that the intercept corresponded to the middle of the data. For the same reason, the sex of participants was coded as female ($-0.310$) or male ($0.690$). Characteristics of the participants (AQ, vocabulary scores) were $z$-transformed.

Participant and item (target word or disambiguating word) were treated as random factors. Following *Barr et al. (2013)*, we adopted "maximal" random factor structures, with random intercepts, slopes, and interactions as appropriate (i.e., "for the highest-order combination of within-unit factors subsumed by each interaction"; *Barr, 2013*, pp 1).

Outliers were removed using a model-based approach, whereby data points with a residual outside of $\pm 2.5$ SD were excluded and the analysis repeated (*Baayen & Milin, 2010*). Quantile–quantile plots were used to confirm a normal distribution of residuals.

For each analysis, we initially used a relatively simple fixed effects model in which $z$-transformed AQ score was allowed to interact with the fixed factor of interest (predictability or ambiguity). When effects of interest were found, we then repeated analyses adding other participant characteristics (age, sex, vocabulary) to the model in order to determine whether they moderated the effect of interest.

**Table 1** Fixed effects in the analysis of predictability effects.

|  | Estimate | Std error | *T* value | *P* value |
|---|---|---|---|---|
| **Initial model** | | | | |
| Intercept | 2.33041 | 0.009216 | | |
| Predictability | −0.025763 | 0.009697 | −2.66 | .008 |
| AQ | 0.015343 | 0.006905 | 2.22 | .026 |
| Predictability × AQ | −0.001227 | 0.008557 | −0.14 | .889 |
| **Optimal model** | | | | |
| Intercept | 2.33051 | 0.008707 | | |
| Predictability | −0.028045 | 0.009791 | −2.86 | .004 |
| Trial | −0.020873 | 0.006423 | −3.25 | .001 |
| AQ | 0.013180 | 0.006480 | 2.03 | .042 |
| Sex | −0.034904 | 0.015305 | −2.28 | .023 |
| Vocabulary | −0.020539 | 0.007333 | −2.80 | .005 |

## RESULTS

### Predictability manipulation

According to the central coherence hypothesis, individuals with high autistic traits should benefit less from a sentence context that makes the target word more predictable. In other words, there should be an interaction between the size of the predictability effect and autistic traits. To test this hypothesis, we used a relatively simple model in which first fixation duration was determined by the interaction of target predictability and $z$-transformed AQ score.

```
1. LogFirstFixationDuration ~ Predictability * zAQ + (1 +
   Predictability | SubjectID) + (1 + Predictability * zAQ |
   TargetWord)
```

Somewhat surprisingly, the main effect of Predictability narrowly failed to achieve significance, $t = -1.92$, $p = .055$. However, inspection of the random effects revealed that one target word, "ditch" was a significant outlier with a strong predictability effect in the unexpected direction. Analyses were therefore repeated excluding trials involving this target word (see Table 1 and Fig. 1). There was now a significant effect of predictability, with predictable target words being fixated for less time than unpredictable targets (effect size: 13 ms). Unexpectedly, there was a significant effect of AQ score with high AQ scores being associated with longer fixation times (effect size 7 ms per SD). However, contrary to predictions of the central coherence hypothesis, there was no hint of an interaction between predictability and AQ score.

Further analyses were conducted in which age, sex, vocabulary, and trial number were added to the model in varying combinations. However, in all of the models, the predictability by AQ interaction remained non-significant. Model comparison (using the anova function in R) suggested the following as the optimal model.

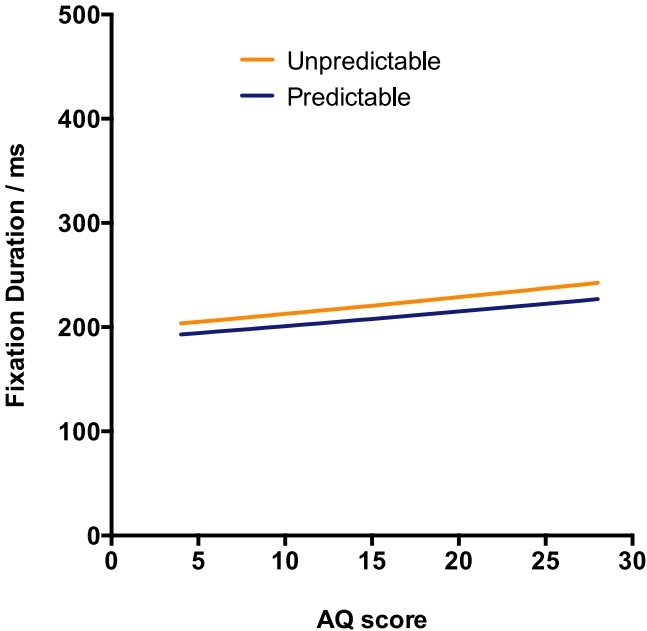

**Figure 1** First fixation duration on the target word under the predictability manipulation.

```
2. LogFirstFixationDuration ~ Predictability + TrialNumber +
   zAQ + Sex + Vocabulary + (1 + Predictability + TrialNumber
   | SubjectID) + (1+ Predictability + zAQ + Sex + Vocabulary |
   TargetWord)
```

As before, target words were fixated for significantly less time in the predictable condition (effect size 14 ms). There was also a significant reduction in fixation time across trials (21 ms from first to last trial). Fixation durations were significantly shorter for females (17 ms), for participants with high vocabulary scores (10 ms per SD), and for those with low AQ scores (6 ms per SD).

## Ambiguity manipulation

Based on previous studies, we expected that participants would spend longer fixating on a disambiguating word that forced them to reinterpret the meaning of an earlier homograph. The "comprehension monitoring" account predicted this effect would be reduced amongst individuals with high levels of autistic traits who should be less likely to notice and attempt to repair any miscomprehension. As our main objective was to investigate individual differences in effect size, we report here the analyses based on the first run dwell time, which gave the clearest effects of condition.

The initial model (Model 3) we employed included ambiguity and AQ scores as interacting fixed effects. The model also included random intercepts and slopes (ambiguity effects) for subjects. For items (target homographs), we included random intercepts and slopes for both ambiguity and AQ as well as a random ambiguity by AQ interaction.

**Table 2 Fixed effects in the analysis of ambiguity (homograph) effects on first run dwell times for the disambiguating word.** Separate analyses were conducted for homophonic and heterophonic homographs.

|  | Estimate | Std error | T value | P value |
|---|---|---|---|---|
| **Homophonic homographs** |  |  |  |  |
| Intercept | 2.410683 | 0.013960 |  |  |
| Ambiguity | 0.009856 | 0.010725 | 0.92 | .358 |
| AQ | 0.012909 | 0.009544 | 1.35 | .177 |
| Ambiguity × AQ | 0.001535 | 0.008290 | 0.19 | .849 |
| **Heterophonic homographs** |  |  |  |  |
| Intercept | 2.41887 | 0.04647 |  |  |
| Ambiguity | 0.02283 | 0.03723 | 0.61 | .542 |
| AQ | 0.03301 | 0.01338 | 2.47 | .014 |
| Ambiguity × AQ | 0.08044 | 0.02412 | 3.34 | <.001 |

```
3.  LogFirstRunDwellTime ~ Ambiguity * zAQ + (1 + Ambiguity |
    SubjectID) + (1 + Ambiguity * zAQ | Homograph)
```

As expected, dwell times on the disambiguating words were longer when they followed a homograph compared to control words. However, this effect fell well short of significance, $t = 1.32, p = .187$. The effects of AQ score, $t = 1.60, p = .110$, and the interaction between ambiguity and AQ score, $t = 1.61, p = .107$, were also non-significant, with the interaction trending in the opposite direction to that predicted by the comprehension monitoring account.

Given that our stimuli included a mixture of homophonic and heterophonic homographs, we conducted further exploratory analyses, coding whether or not the homograph in the homograph-control pair was homophonic (Model 4).

```
4.  LogFirstRunDwellTime ~ Ambiguity * Homophony * zAQ + (1 +
    Ambiguity * Homophony | SubjectID) + (1 + Ambiguity * zAQ |
    Homograph)
```

This reanalysis revealed a highly significant three-way interaction between ambiguity, homophony, and AQ score, $t = -3.48, p < .001$. We therefore re-examined the data for homophonic and heterophonic homographs separately (using Model 3) (see Table 2 and Fig. 2). For homophonic homographs, there was no effect of ambiguity, no effect of AQ, and no interaction between ambiguity and AQ.

For heterophonic homographs, there was again no main effect of ambiguity, but there was a significant effect of AQ score and a significant interaction such that high AQ scores were associated with a larger (more positive) ambiguity effect—that is, in the opposite direction to predictions. Given that there are only five heterophonic homographs, we repeated the analyses excluding each homograph in turn. However, the pattern of results was identical in each case, indicating that the interaction was not driven by any single homograph.

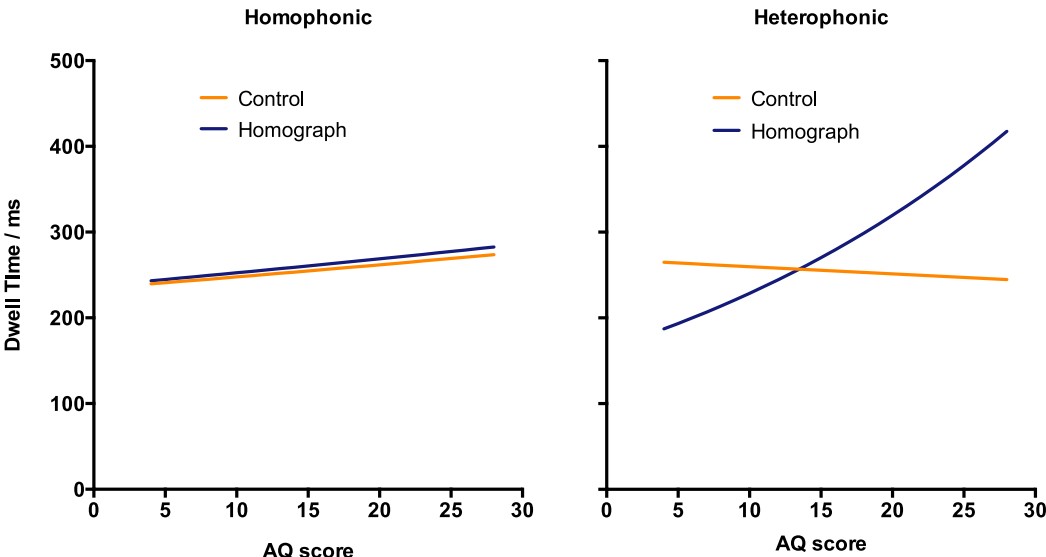

**Figure 2  First run dwell time on the disambiguating word under the ambiguity manipulation.** Results are plotted separately for sentences containing homophonic and heterophonic homographs.

Finally, given the failure to find a significant main effect of ambiguity a series of exploratory analyses were conducted using a range of alternative eye-tracking measures as the dependent variable. These included first fixation duration, go past duration (the time from fixating on the disambiguating word to fixating on the subsequent word, allowing for regressions to earlier words), first run duration on the region encompassing the disambiguating word and the next word in the sentence, and the time from fixating on the target word to fixating on the last word in the sentence. None of these extra analyses revealed a significant ambiguity effect.

## DISCUSSION

There is now a growing body of evidence to suggest that cognitive strengths and weaknesses associated with autism may also be found amongst individuals in the general population who show high levels of autistic traits. Given the poor performance of autistic individuals on tests of homograph reading, we predicted similar difficulties would be experienced by adults with relatively high levels of autistic traits. The eye tracking test devised for this study allowed us to go beyond previous studies and examine two competing explanations of homograph reading difficulty in autism—a reduced influence of prior context (weak central coherence) and a failure of comprehension monitoring. However, neither of these accounts received support.

According to the weak central coherence account, individuals with autism tend to process words out of context. Thus we predicted that high autistic traits should be associated with insensitivity to preceding sentence context, measured with respect to gaze time on the target word. While we did find the expected main effect of predictability, there was no hint of an interaction with AQ scores, and thus no support for our hypothesis. One interpretation of this finding is that lack of context sensitivity is not in fact a characteristic

of autism and thus should not be expected in association with autistic traits. Our findings are thus consistent with the numerous studies using tasks other than homograph reading that have failed to find an autism-specific reduction in context sensitivity. However, until we collect data from clinically diagnosed individuals with autism using the current task, it is impossible to exclude an alternative interpretation—that individuals with autism experience reduced context sensitivity but this does not extend to non-autistic individuals with high levels of autistic traits.

Our alternative explanation for homograph reading difficulties faired no better than the central coherence account. We had hypothesized that, like non-autistic children with reading comprehension problems, participants with autism fail to monitor for errors of comprehension during reading. Therefore, we predicted that participants would spend longer fixating on words that required them to revise their initial (incorrect) interpretation of a homograph, but that this effect would be reduced in participates with higher AQ scores. Again this prediction was not supported, with no interaction between AQ score and condition.

An important point to note here is that the main effect of ambiguity (homograph vs. control) did not achieve statistical significance. Thus, a reasonable interpretation of our findings is simply that the ambiguity manipulation was unsuccessful and the lack of an interaction with AQ score is, therefore, difficult to interpret. Our design was motivated by previous studies involving homographs that are disambiguated later in the sentence. However, there are some subtle but potentially important differences in the stimulus construction and analysis. Specifically, our sentences were carefully designed such that there was a single word that clearly disambiguated the homograph, allowing us to look at fixations on the disambiguating word itself. This contrasts with previous studies, in which a disambiguating *clause* was inserted after the homograph and reading time was operationalized as the time from first fixating within the disambiguating region to the time a button was pressed to move on to the next sentence. Arguably, ours is a tighter and more controlled design. Our null result for the ambiguity manipulation indicates that participants do not necessarily attempt to resolve any ambiguity as soon as they encounter a word that is inconsistent with their initial interpretation. It may be that this process takes place long after the participants' eyes have moved on from the disambiguating word, and perhaps only after they have completed the sentence.

That being said, closer inspection revealed an intriguing three-way interaction, whereby the interaction between the size of the ambiguity effect and AQ scores was itself moderated by homophony—that is, whether the two meanings of the homograph had the same pronunciation or not. For homophonic homographs, there was no effect of ambiguity and no interaction with AQ scores. In contrast, for heterophonic homographs, there was a significant interaction between ambiguity and AQ scores, but this was in the opposite direction to predictions, with high AQ scores being associated with a larger rather than a smaller ambiguity effect. One possibility is that, at least for some individuals, the (as it happens incorrect) phonological memory representation of the preceding homograph prompts an immediate attempt to resolve the ambiguity. It is perhaps notable here that

individuals with higher AQ scores also tended to have relatively longer fixation times regardless of sentence type or condition. This slower and perhaps more deliberate reading style might allow these participants to register the incongruity between the disambiguating word and the preceding homograph even before they have saccaded to the next word in the sentence.

Clearly this account of our data is speculative and there are a number of important caveats. First, there were only five heterophonic homographs and the counterbalancing design entailed that each participants only received two or three of these (with the other corresponding sentences appearing in the control condition). Second, although significant, the three-way interaction between group, homophony, and ambiguity was part of an exploratory post hoc analysis. These findings would have to be replicated, ideally in an orthography such as Hebrew that has many more heterophonic homographs than English, before drawing any strong conclusions.

In summary, while providing a tantalizing suggestion of differences in reading style associated with subclinical autistic traits, the main outcome of the current study is a lack of support for either the weak central coherence account or our alternative "comprehension monitoring" account. Although the findings from the ambiguity manipulation are open to several interpretations, the results from the predictability manipulation provide clear evidence *against* the proposal that high levels of autistic traits are associated with reduced sensitivity to sentence context.

Despite the oft-repeated claim that individuals with autism are insensitive to sentence context, this is, to our knowledge, the first investigation of sentence context effects in relation to autistic traits. In perhaps the closest existing study, *Stewart & Ota (2008)* reported that high levels of autistic traits were associated with a reduction in the Ganong effect, whereby perception of an ambiguous phoneme (e.g., the sound between /g/ and /k/) is affected by its lexical context. It is important to note that our study had considerably more participants (71 vs. 51) and used a task that was conceptually closer to those used in autism research, targeting sentence-rather than lexical-level context effects. Stewart and Ota claimed support for the weak central coherence account—and for its extension into the non-autistic population. However, a cited reference search indicates that there have subsequently been no published studies investigating the assumption that the Ganong effect will also be reduced in individuals with autism.

In this context, we believe it is important to publish the failures to find significant associations with autistic traits as well as the "successes". Indeed, studies investigating subclinical autistic traits may be particularly susceptible to publication bias. Statistically significant associations provide the compelling narrative that "everybody is a little bit autistic" and are relatively straghtforward to publish, often in high impact journals. In contrast, a null result may be easily dismissed because the study did not involve *bona fide* individuals with autism, because the study is considered underpowered, or simply of lesser interest. In our view, it is only by gaining a complete picture of all results that researchers will be able to determine how and to what extent the characteristics of autistic individuals extend into the typical population.

### Funding

Jon Brock was supported by an Australian Research Council Australian Research Fellowship (Grant DP098466). He is a Chief Investigator at the Australian Research Council Centre of Excellence in Cognition and its Disorders (Grant CE110001021). The funders had no role in study design, data collection and analysis, decision to publish, or preparation of the manuscript.

### Grant Disclosures

The following grant information was disclosed by the authors:
Australian Research Council Discovery Project: DP098466.

### Competing Interests

Jon Brock is an Academic Editor for PeerJ. The authors declare there are no competing interests.

### Author Contributions

- Nathan Caruana conceived and designed the experiments, performed the experiments, analyzed the data, wrote the paper, prepared figures and/or tables, reviewed drafts of the paper.
- Jon Brock conceived and designed the experiments, analyzed the data, wrote the paper, prepared figures and/or tables, reviewed drafts of the paper.

### Human Ethics

The following information was supplied relating to ethical approvals (i.e., approving body and any reference numbers):

The study was approved by the Macquarie University Human Research Ethics Committee (Ref D00167).

### Supplemental Information

Supplemental information for this article can be found online at http://dx.doi.org/10.7717/peerj.466.

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
