# Peer review of "No association between autistic traits and contextual influences on eye-movements during reading"

_PeerJ, doi:10.7717/peerj.466_

## Round 0.1 · original submission · Major Revisions

Dear Authors,

The comments of both peer reviewers are enclosed. Please do revise the manuscript especially to the comments of the second peer reviewer.

Thanking you.

·

Basic reporting

The Introduction is well-written, provides good coverage of relevant background literature, and nicely motivates the study by connecting it to broad theoretical issues in the study of autism. The materials and procedures are reported in sufficient detail to enable replication. The data analysis procedures were also clearly and fully reported. Although the reporting of the results is generally good, the authors need to include more statistical information to better enable readers to interpret the results.

The tables and text reporting inferential statistics give only the test statistic value, but no p-value. The authors should also report a (two-tailed) p-value for each test statistic. This can be obtained using the R function 'pval <- 2*(1-pnorm(x))' where x is the test statistic. For instance, 2*(1-pnorm(1.96)) yields a p value of .05.

It would also be useful to give the reader some indication of the magnitude of the differences on the natural scale of the DVs rather than merely reporting only the test statistics and the direction of the effect. I am not requesting effect size measures, as these are difficult to calculate for mixed-effects models, just basic reporting of mean differences, e.g., for the effects of predictability, gender, etc. For variables that are continuous, statements such as, "for every 10 AQ points, the duration increased on average by Y ms" would be useful.

The model formulae for predictability (lines 203-205, and lines 220-222) include the predictors "Predictability" and "Context", the later appearing in the by-item random effects term. The "Context" variable is never defined, and I would guess that it is just another name for the Predictability variable, as Predictability should appear in the by-item random effects. The authors need to either fix the typo or describe the Context variable and explain why it only appears in the by-item random effects.

Experimental design

The study is appropriately designed and carefully executed. I have no concerns with this aspect of the study.

Validity of the findings

The authors are to be commended for using a state-of-the-art mixed-effects modeling approach for their analyses, which allows for the proper treatment of the AQ as a continuous variable, as well as for the inclusion of control variables as covariates. The manuscript clearly distinguishes planned confirmatory analyses from unplanned exploratory analyses. The authors are refreshingly candid about the possible limitations of their study, including the possible inefficacy of the ambiguity manipulation.

Additional comments

Here are some minor issues to fix:

[numbers at the beginning of a line preceded by a colon refer to the line number(s) in the manuscript]

26: Frith 1989 not in reference list

45-46: scope ambiguity; "autistic individuals may not recognize when the sentence stops making sense because they have misconstrued the homograph" Here, the clause following 'because' is given as a reason why the sentence stops making sense, but the way the sentence is constructed, it could be misread as offering a reason for their failure to recognize when the sentence stops making sense.

110: missing & misplaced bracket in citation

113-138: by 'forms', do you mean stimulus lists? forms is confusing

173: Please report which version of lme4 you used [can find version number by loading library(lme4) followed by a call to sessionInfo()]. May be important as different versions could return different results

173: for the 'R' citation, cite the R Core Team rather than Baayen, Davidson & Bates 2008 (the citation for your specific version of R can be obtained using the citation() command). Citation of Baayen Davidson & Bates would be better placed at the end of line 181.

176: "was also coded as a fixed factors"

179: one level of the gender predictor is missing a negative sign, needed to interpret parameter estimates

188: "mle4"

·

Basic reporting

Is the main difference between the weak coherence and the comprehension monitoring explanations in the timing of the error detection, which is either a result of predictive mechanisms (the weak coherence account) vs. post-dictive mechanisms (the comprehension monitoring) ? This distinction has been previously made with respect to mental state understanding, in autism (e.g. Senju, 2012) and it would be interesting to know whether the authors believe it to be relevant to their work.

Experimental design

Most previous research seems to have been carried out using verbal responses. Why do the authors choose to employ eye-tracking ? Understanding this methodological choice is particularly important in view of the null results.

It is not clear why certain effects were treated as fixed, other as random, and there seem to be some contradictions (e.g. l. 235-238 the interaction term is included as a fixed or random effect ?)

Validity of the findings

Although I am sympathetic to the publication of null results, I think they are most valuable when they emerge from studies employing well-established paradigms/measures that had shown positive results in previous studies/other populations, which the current findings would contradict. The authors themselves acknowledge that the measure used (fixation duration on target words) and the paradigms (modified versions of published paradigms) may not have been the most appropriate (l. 295- 307). If there are better measures than i'd like to see them employed. A main effect of the experimental variable is at least expected in the control group (not the case with the ambiguity effects). Not knowing whether autistic individuals behave differently in these exact these task, it is difficult to know what to expect from individuals with sub-clinical autistic traits.

Additional comments

To understand the implications of these null findings this works needs validation with either more widely used paradigms or with a population of individuals with ASD.

---

## Round 0.2 · accepted · Accept

Thank you for the submission the revised manuscript which is now accepted for publication after re-review.Congratulations!

·

Basic reporting

This is a review of a revised version of a previous manuscript. In my last review, my main concern was that there was insufficient statistical reporting. The current version shows dramatic improvement in this area. I believe no further changes are needed.

Experimental design

No Comments

Validity of the findings

No Comments

Additional comments

A very nice contribution to the literature.